# Combined Live Oral Priming and Intramuscular Boosting Regimen with Rotarix^®^ and a Nanoparticle-Based Trivalent Rotavirus Vaccine Evaluated in Gnotobiotic Pig Models of G4P[6] and G1P[8] Human Rotavirus Infection

**DOI:** 10.3390/vaccines11050927

**Published:** 2023-05-02

**Authors:** Casey Hensley, Charlotte Nyblade, Peng Zhou, Viviana Parreño, Ashwin Ramesh, Annie Frazier, Maggie Frazier, Sarah Garrison, Ariana Fantasia-Davis, Ruiqing Cai, Peng-Wei Huang, Ming Xia, Ming Tan, Lijuan Yuan

**Affiliations:** 1Department of Biomedical Sciences and Pathobiology, Virginia-Maryland College of Veterinary Medicine, Virginia Tech, Blacksburg, VA 24060, USA; 2INCUINTA, Instituto de Virología e Innovaciones Tecnológicas (IVIT), Instituto Nacional de Tecnología Agropecuaria (INTA)-CONICET, Buenos Aires C1033AAE, Argentina; 3Division of Infectious Diseases, Cincinnati Children’s Hospital Medical Center, Cincinnati, OH 45229, USA; 4Department of Pediatrics, University of Cincinnati College of Medicine, Cincinnati, OH 45229, USA

**Keywords:** rotavirus nanoparticle, gnotobiotic pig, priming immunization

## Abstract

Human rotavirus (HRV) is the causative agent of severe dehydrating diarrhea in children under the age of five, resulting in up to 215,000 deaths each year. These deaths almost exclusively occur in low- and middle-income countries where vaccine efficacy is the lowest due to chronic malnutrition, gut dysbiosis, and concurrent enteric viral infection. Parenteral vaccines for HRV are particularly attractive as they avoid many of the concerns associated with currently used live oral vaccines. In this study, a two-dose intramuscular (IM) regimen of the trivalent, nanoparticle-based, nonreplicating HRV vaccine (trivalent S_60_-VP8*), utilizing the shell (S) domain of the capsid of norovirus as an HRV VP8* antigen display platform, was evaluated for immunogenicity and protective efficacy against P[6] and P[8] HRV using gnotobiotic pig models. A prime–boost strategy using one dose of the oral Rotarix^®^ vaccine, followed by one dose of the IM trivalent nanoparticle vaccine was also evaluated. Both regimens were highly immunogenic in inducing serum virus neutralizing, IgG, and IgA antibodies. The two vaccine regimens failed to confer significant protection against diarrhea; however, the prime–boost regimen significantly shortened the duration of virus shedding in pigs challenged orally with the virulent Wa (G1P[8]) HRV and significantly shortened the mean duration of virus shedding, mean peak titer, and area under the curve of virus shedding after challenge with Arg (G4P[6]) HRV. Prime–boost-vaccinated pigs challenged with P[8] HRV had significantly higher P[8]-specific IgG antibody-secreting cells (ASCs) in the spleen post-challenge. Prime–boost-vaccinated pigs challenged with P[6] HRV had significantly higher numbers of P[6]- and P[8]-specific IgG ASCs in the ileum, as well as significantly higher numbers of P[8]-specific IgA ASCs in the spleen post-challenge. These results suggest the promise of and warrant further investigation into the oral priming and parenteral boosting strategy for future HRV vaccines.

## 1. Introduction

Human rotavirus (HRV) remains the leading cause of childhood dehydrating diarrhea in low- and middle-income countries (LMICs), despite the use of live attenuated oral HRV vaccines [1,2,3,4]. The two most used, commercially available vaccines are the monovalent Rotarix^®^ (G1P[8]) and pentavalent RotaTeq^®^ (G1-4P[8]). These vaccines, though highly effective in developed countries, do not offer the same protection for children in LMICs for various reasons, mainly associated with the gastrointestinal (GI) tract [1,2,5,6,7]. Some of these issues include malnutrition, circulating maternal antibodies, concurrent use of other oral vaccines, and gut dysbiosis [8,9,10,11,12]. Malnutrition and gut dysbiosis have been associated with reduced immune response to live oral attenuated rotavirus vaccines previously [6,8,13,14]. Furthermore, there is evidence that the use of live oral poliovirus vaccines, which are often given in conjunction with HRV vaccines, can inhibit HRV vaccine replication, further reducing vaccine take [8,10,15]. Additionally, P[4] and P[6] HRVs are commonly co-circulating with P[8] strains, highlighting the need for vaccines offering cross-P-type protection, especially as new strains (P[6] in particular) begin to emerge [16,17,18]. Furthermore, these vaccines are associated with a small risk of intussusception [1,19,20]. These reasons, along with the massive yearly economic burden of HRV infection, make parenteral routes of HRV vaccination particularly attractive in LMICs where the HRV burden is the highest [1,21]. HRV is an enteric pathogen that replicates in the small intestine, and the generation of HRV-specific local IgA and virus-specific IFN-γ+ T cell responses are associated with protection from the disease in Gn pigs [22]. Because of this, live oral attenuated vaccines have been the focus of vaccine development in the past, due to the vaccines’ ability to replicate in the gut and induce similar immune responses to natural HRV infection without inducing severe disease. On the contrary, parenteral HRV vaccines have been shown to be highly immunogenic and are now the focus of current efforts to address live oral vaccines’ shortcomings [23]. Unfortunately, a phase III human trial for a parenteral HRV vaccine was recently terminated due to a lack of improved protective efficacy as compared to currently used live oral attenuated vaccines [24]. Despite this, parenteral vaccines still may have an important role in protecting against disease. Priming and boosting regimens for HRV vaccination have been tested in a gnotobiotic (Gn) pig model successfully and have often shown superior results compared to repeated immunization with the same vaccine. In a group of studies that investigated various prime–boost strategies in Gn pigs, oral attenuated HRV as a priming immunization (analogous to Rotarix^®^) and both intranasal (IN) adjuvanted VLPs or intramuscular (IM) DNA plasmid as boosting doses induced higher protection than any other regimen [23].

HRV binding to host cells is mediated by the VP4 protein, which is proteolytically cleaved in the host’s gut. This cleavage yields the proximal VP5* and distal VP8*, the latter of which is responsible for facilitating binding and entry, as well as being a major target for host immune responses [25,26]. Because VP8* has been shown to induce virus-neutralizing (VN) antibody responses in several studies, it has remained the target for HRV vaccine development [27,28,29,30]. In this study, a nanoparticle-based nonreplicating HRV vaccine, utilizing the shell (S) domain of the capsid protein (VP1) of norovirus (NoV) as a platform, was developed. The S domain of NoV self-assembles into a 60-valent icosahedral nanoparticle with exposed C-termini, which are amenable to the insertion of foreign proteins [31,32]. In this vaccine, VP8* from the three most predominant HRV P-types (P[4], P[6], and P[8]) were fused to the exposed C-termini, forming the S_60_-VP8* trivalent nanoparticle vaccine. This vaccine is easily produced in high quantities using an *E. coli* expression system and has been shown to be highly immunogenic in murine models [33,34].

The goal of this study was to evaluate the safety, immunogenicity, and protective efficacy of the trivalent S_60_-VP8* vaccine in Gn pigs as both a two-dose IM regimen and as a booster dose preceded by an oral priming dose using the commercially available Rotarix^®^. Gn pigs have long been used for vaccine efficacy studies as they develop a similar disease to that of humans after HRV inoculation, including intestinal pathology [23]. They are also immunologically similar to humans, and their gnotobiotic status allows for minimal immune-related variables when evaluating vaccine efficacy [23]. In this study, we utilized the well-established Gn pig model of P[8] HRV infection and diarrhea as well as the newly characterized P[6] HRV infection and diarrhea model [16]. The inclusion of both models allows for the generation of a deeper understanding of potential cross-protection induced by candidate HRV vaccines. This is important as the incidence of P[6] HRVs have been on the rise in recent years and continue to circulate globally, but currently licensed vaccines do not contain P[6] immunogens [17].

## 2. Materials and Methods

### 2.1. Rotaviruses for Challenge and Immunoassays

The virulent Wa HRV (G1P[8]) and Arg HRV (G4P[6]) inocula consisted of pools composed of small intestinal contents collected from the 29th passage and 6th passage of the viruses in neonatal Gn pigs, respectively. A dose of 10^5^ focus-forming units (FFU) of Wa and Arg HRV diluted in 5 mL of Diluent #5 (minimal essential media [ThermoFisher Scientific, Waltham, MA, USA]; 100 IU of penicillin per mL, 0.1 mg of dihydrostreptomycin per ml; and 1% HEPES) was used for the oral challenge of Gn pigs. The median infectious and diarrhea doses of both Wa and Arg HRV were determined previously to be approximately 1 FFU, and 10^5^ FFU was the optimal challenge dose that will induce diarrhea and virus shedding in 100% of unimmunized Gn pigs [16,35]. The supernatant of attenuated Wa HRV-infected African green monkey kidney MA104 cells (ATCC CRL-2378.1™) was semi-purified by centrifugation and used as a positive control in antigen enzyme-linked immunosorbent assay (ELISA) and cell culture immunofluorescence (CCIF).

### 2.2. Vaccine

Three P-types of S_60_-VP8* nanoparticles (P[8], P[6], and P[4]) were produced and purified as previously described [31,33] and mixed as the trivalent vaccine in the liquid formulation in phosphate buffer saline (PBS). The individual nanoparticles and the trivalent vaccine were assessed by (1) SDS-PAGE for protein quality; (2) gel filtration chromatography and EM inspection for nanoparticle formation and structural integrity [31,33]. Bacterial endotoxin was removed by using ToxinEraser™ Endotoxin Removal Resin (GenScript, Cat. No L00402. Piscataway, NJ, USA). The vaccines were formulated by mixing each S_60_-VP8* P[8], P[6], and P[4] or S_60_ alone with 600 µg/dose of aluminum hydroxide adjuvant (Alhydrogel, InvivoGen, San Diego, CA, USA). The vaccine dose was originally designed to contain 200 µg of each P-type. However, due to the need for filtration to eliminate contaminating bacteria, the actual amount of antigen in the vaccine was reduced. The actual amount of antigens in the vaccine measured with Nanodrop was 94 µg of P[4], 52 µg of P[6], and 143 µg of P[8]. Sterility of antigens was confirmed using blood agar and thioglycolate broth prior to inoculation of Gn pigs.

### 2.3. Vaccine Inoculation, Virus Challenge, and Sample Collection of Gn Pigs

Piglets were derived via hysterectomy of near-term sows and maintained in germ-free isolators for the duration of the study [36]. Sterility was verified by weekly rectal swabs (RS) on blood agar and in thioglycolate broth. Pigs were fed ultra-high temperature (UHT) treated sterile whole cow’s milk (The Hershey Company, Hershey, PA, USA) for the duration of the study. Pigs were inoculated with either the priming dose of trivalent S_60_-VP8* IM (group 1) or Rotarix^®^ orally on post-partum day (PPD) 5 (also referred to as post-vaccination day (PVD) 0) and subsequently received the same dose IM boosting of trivalent S60-VP8* at PVD 14 (group 2) (Figure 1). Control pigs (group 3) were immunized on the same schedule with two IM administrations of the S_60_ nanoparticle without VP8* antigen. Serum was collected before each vaccination at PVD 0, PVD 14, and PVD 28/post-challenge (PCD) 0 and at euthanasia (PCD 7) to evaluate P[8], P[6], and P[4] VP8*-specific IgG and IgA by direct ELISA and virus neutralizing (VN) titers by virus neutralization assay.

Pigs were assigned randomly to three groups per challenge strain (Figure 1) for a total of 45 combined. Pigs were orally challenged on PVD 28/PCD 0 with 1 × 10^5^ FFU of either virulent Wa HRV (G1P[8]) or virulent Arg HRV (G4P[6]) and monitored for diarrhea and virus shedding via daily RS from PCD 0–7. All pigs were fed 4 mL 200 mM sodium bicarbonate 10 min prior to the HRV challenge to neutralize stomach acidity. Pigs were euthanized at PCD 7. At euthanasia, small intestinal contents (SIC) and large intestinal contents (LIC) were collected and processed as previously described, for the detection of intestinal antibody responses and viral antigen presence by direct and sandwich ELISA, respectively, as well as infectious virus particle counting by CCIF. Spleen, ileum, and whole blood were collected for extraction of MNCs to be used in the detection of P[8] and P[6] VP8*-specific IgG and IgA ASC responses by enzyme-linked immunosorbent spot (ELISpot) and P[8], P[6], and P[4]VP8*-specific IFN-γ-producing CD4+ and CD8+ T cells by flow cytometry [37,38].

### 2.4. Assessment of Diarrhea and Detection of Fecal Virus Shedding by Antigen ELISA and Virus CCIF

To assess the severity of diarrhea, fecal consistency was recorded, using the RS taken from PCD 0 to 7, as 0: solid, 1: pasty, 2: semi-liquid, and 3: liquid, with any score ≥2 being considered diarrhea [35]. Antigen ELISA was used for the detection of HRV VP6 and CCIF for the detection of infectious virus particles, as previously described [35,39].

### 2.5. Detection of HRV VP8*-Specific Serum and Intestinal IgA and IgG Antibody by ELISA

Anti-HRV VP8* IgA and IgG antibody titers in serum and intestinal contents were measured by using an isotype-specific ELISA as described previously [34], using recombinant GST-VP8* protein of P[8], P[6], or P[4] type as capture antigens that were coated to plates at a concentration of 1.5 µg/mL overnight at 4 °C. Both serum and intestinal contents were clarified by centrifugation at 10,000 rpm for 10 min. Coated plates were blocked with 5% non-fat milk in PBS at 37 °C, followed by incubation with four-fold serially diluted serum sample (starting from 1:4 to 1:1,048,576) or intestinal content sample (starting from 1:4 to 1:65,536) overnight at 4 °C. The bound antibodies were detected using goat anti-pig IgA antibody conjugated with horseradish peroxidase (HRP) (Bethyl Laboratories, Inc., Montgomery, TX, USA) or goat anti-pig IgG heavy/light chain antibody conjugated with HRP (Bethyl Laboratories, Inc.) at 1:8000 diluted in PBST with 1% non-fat milk. Color was developed using TMB Substrate Reagent Set (BD Biosciences), and optical density (OD) was read at 450 nm.

### 2.6. Flow Cytometry for Detection of IFN-γ-Producing CD3+CD4+ and CD3+CD8+ T Cells

Mononuclear cells isolated at PCD 7 from peripheral blood (PCD 7), ileum, and spleen were diluted to a concentration of 1 × 10^6^ cells/mL and seeded into 12-well plates [38]. Cells were re-stimulated for 17 h at 37 °C in 5% CO_2_ with one of seven antigens: (1) medium (negative control), (2) PHA (10 µg/mL; positive control), (3) P[4] protein (12 µg/mL), (4) P[6] protein (12 µg/mL), and (5) P[8] protein (12 µg/mL). Anti-CD49d mAb (0.5 µg/mL of clone 9F10, catalog #555502, BD Biosciences, Franklin Lakes, NJ, USA) was added to all samples before incubation for co-stimulation. After 12 h, Brefeldin A (5 µg/mL) was added for 5 h at 37 °C in 5% CO_2_. After the total 17 h incubation, cells were washed with 2 mL of commercial stain buffer and transferred to 5 mL Falcon round-bottom polypropylene tubes for 8 min centrifugation at 800× *g* at 4 °C, followed by discarding of the supernatant. Primary antibodies used for staining were mixed in a trivalent that included FITC-conjugated mouse (IgG2b) anti-pig CD4α (clone 74-12-4, catalog #559585, BD Biosciences, Franklin Lakes, NJ, USA), SPRD-conjugated mouse (IgG2a) anti-pig CD8α (clone 76-2-11, catalog #4520-13, Southern Biotech, Birmingham, AL, USA), mouse (IgG1) anti-pig CD3ε (clone PPT3, catalog #4510-01, Southern Biotech, Birmingham, AL, USA), and GloCell^TM^ Fixable Viability Dye Violet 450 (catalog #75009, StemCell Technologies, Cambridge, MA, USA) in 100 µL of stain buffer per sample. Samples were incubated with the trivalent for 15 min at 4 °C and subsequently washed with 500 µL of wash buffer, followed by 8 min centrifugation at 800× *g* at 4 °C. Then, secondary antibody APC-conjugated rat (IgG1) anti-mouse for CD3ε (clone X56, catalog #550874, BD Biosciences, Franklin Lakes, NJ, USA) diluted in 100 µL of stain buffer was added and incubated for another 15 min at 4 °C. The washing and centrifugation step was repeated, then, 100 µL of BD Cytofix/Cytoperm (cat #554714, BD Biosciences, Franklin Lakes, NJ, USA) permeabilizing/fixation solution containing 4.2% formaldehyde was added into each sample and incubated for 30 min at 4 °C. Washing and centrifugation were then repeated. PE-conjugated mouse (IgG1) anti-pig IFN-γ (clone P2G10, catalog #559812, BD Biosciences, Franklin Lakes, NJ, USA) diluted in 100 µL of stain buffer was added and incubated for another 30 min at 4 °C. A last washing step with 2 mL of stain buffer was performed, followed by centrifugation. Cells were resuspended in 250 µL of stain buffer and stored away from light at 4 °C until delivery to Virginia Tech’s flow cytometry core for analysis within 24 h for acquisition on a BD FACSArial^TM^ II flow cytometer.

### 2.7. Detection of VP8*-Specific Antibody-Secreting Cells by ELISpot Assay

Plates (96-well Falcon [Corning], Corning, NY, USA) were coated with 50 µL P[8], P[6], or P[4] at the concentration of 5 µg/mL in 50 mM carbonate coating buffer (pH 9.6). Plates were incubated at 37 °C for 30 min, then stored at 4 °C overnight. Plates were washed twice with PBST (pH 7.4, with 0.05% Tween 20), blocked with 100 µL/well 4% BSA in PBS (pH 7.4), and incubated at 37 °C for 1 h. Plates were washed three times with ddH_2_0. Separately, extracted MNCs were diluted to single-cell suspension at concentrations of 5 × 10^6^ and 5 × 10^5^ cells/mL with E-RPMI and 100 µL of each diluted cell suspension was added to duplicate wells. Plates were then centrifuged at 500 rpm for 5 min at RT, followed by incubation at 37 °C for 12 h. Plates were then washed five times with PBST, and 100 µL/well biotinylated goat anti-porcine IgA diluted 1:20,000 (Bethyl A100-102B, 1 mg/mL) or IgG (Bethyl A100-104B, 1 mg/mL) diluted 1:20,000 in PBST was added to each well. Plates were incubated at RT for 2 h and washed five times with PBST. To ensure cells were completely washed away, the plates were knocked vigorously on paper towels between each wash. Next, 100 µL/well HRP-conjugated streptavidin diluted 1:30,000 in PBS was added to the plates followed by incubation at RT for 1 h. Plates were washed five times with PBST, and 50 µL/well KPL TrueBlue substrate (ready-to-use substrate from SeraCare, Milford, MA, USA) was added to each well. Plates were then incubated for 1–2 h at RT until spots became dark blue. Spots were counted with ELISpot analyzer S5 (ImmunoSpot, Cleveland, OH, USA) and reported as IgA or IgG ASC per 5 × 10^5^ MNC.

### 2.8. Virus Neutralization Assay

HRV neutralization assays were performed using a fluorescence-based plaque reduction assay as described previously [34]. Briefly, trypsin-treated HRVs of P[8] (Wa strain, G1P[8]), P[6] (ST-3 strain, G4P[6]), and P[4] (DS-1 strain, G2P[4]) types were each incubated with serially diluted serum samples. The HRVs were then added to MA104 cells on 96-well plates. After further culture for 16 h, the plated cells were frozen with 80% (*v*/*v*) acetone and then blocked with non-fat milk. The HRV-infected cells were stained with a 2KD1 nanobody against VP6 labeled with Alexa fluor 488 at a 1/500 dilution [40]. The bound antibodies were detected using FITC-labeled goat anti-guinea pig IgG antibody. Fluorescence plaques were photographed using the Cytation 5 imaging reader and then counted. Neutralization titers were defined as the maximum dilutions of the serum samples showing at least 50% reduction in fluorescence-formation plaques.

### 2.9. Statistical Analysis

Pigs were randomly assigned to treatment groups by animal care staff upon derivation, regardless of sex or body weight. Mean numbers of ASCs in post-mortem tissues in each treatment group were compared using the Kruskal–Wallis non-parametric rank sum test followed by Dunn’s multiple comparisons tests. Daily mean diarrhea scores and CCIF titers were evaluated using two-way ANOVA of repeated measures through time, followed by Tukey’s multiple comparisons tests. All other diarrhea and virus-shedding parameters were evaluated using one-way ANOVA followed by Tukey’s multiple comparisons tests, or Kruskal–Wallis test when homoscedasticity and normality assumptions were not met. Mixed-effects analysis followed by Dunnett’s multiple comparisons test was used for comparing serum and intestinal antibody responses, and VN titers were evaluated using mixed-effects analysis with a two-stage linear step-up procedure. Because only two treatment groups were compared, total numbers and mean frequencies of T cell subsets were evaluated using multiple Mann–Whitney tests. All analyses were performed using GraphPad Prism 9 (GraphPad Software, San Diego, CA, USA). A *p*-value lower than 0.05 was accepted as statistically significant.

## 3. Results

### 3.1. Prime–Boost Regimen Significantly Reduced Virus Shedding in Both Challenge Groups

Gn pigs were challenged with either virulent Wa or Arg HRV at PVD 28/PCD 0, and clinical signs/virus shedding were monitored via daily RS. These data are summarized in Table 1 and Table 2, respectively. All pigs in all treatment groups developed diarrhea and virus shedding after challenge with either Wa or Arg HRV. Although no significant differences were detected, trends for both challenge groups showed lower mean duration days, mean cumulative scores, and AUC of diarrhea for prime–boost vaccinated pigs as compared to controls (Table 1 and Table 2). Significant differences in mean daily diarrhea scores were detected for Wa HRV-challenged prime–boost vaccinated pigs at PCDs 4–6 as compared to controls (Figure 2A). Arg HRV-challenged pigs vaccinated with the prime–boost regimen had lower daily mean diarrhea scores from PCDs 1–5, although they were not statistically significant (Figure 2B).

Protection against virus shedding was more pronounced. In Wa HRV-challenged pigs, the onset of virus shedding was significantly delayed, and the duration of virus shedding was significantly reduced in prime–boost vaccinated pigs as compared to controls (Figure 3A,B). In Arg HRV-challenged pigs, the duration of virus shedding, mean peak titers, and AUC of virus shedding were all significantly reduced in prime–boost vaccinated pigs (Figure 3F–H). In Wa HRV-challenged pigs, daily mean CCIF titers were significantly lower in prime–boost vaccinated pigs than in other groups on PCD 5 (Figure 4A). In Arg HRV-challenged pigs vaccinated with the prime–boost regimen, mean CCIF titers were significantly lower from PCDs 4–6 as compared to controls (Figure 4B).

### 3.2. Both Vaccine Regimens Were Highly Immunogenic and Induced Strong Serum IgG and IgA Responses in Gn Pigs before and after Challenge with Wa or Arg HRV

Serum samples collected at PVD 0, 14, 28, and PCD 7 were used to evaluate vaccine-induced, P-type specific IgG and IgA antibodies in Gn pigs. In both challenge groups, pigs vaccinated with trivalent nanoparticle 2× had significantly higher titers of P[8], P[6], and P[4]-specific serum IgG as compared to controls after just one-dose vaccination (PVD 14), which lasted until PVD 35/PCD 7 (Figure 5A and Figure 6A). Gn pigs in both challenge groups who received the prime–boost regimen developed significantly higher titers for all three P-types as compared to controls at PVD 28, which lasted until PVD 35/PCD 7 (Figure 5A and Figure 6A).

P[8], P[6], and P[4]-specific serum IgA antibody titers in the Wa HRV-challenged group were significantly higher in trivalent nanoparticle 2×-vaccinated pigs at PVD 28/PCD 0 and PVD 35/PCD 7 (Figure 5B). P[6]-specific IgA became significantly higher in trivalent nanoparticle 2×-vaccinated Wa-challenged pigs early, after only one-dose vaccination (PVD 14), and again lasted until PVD 35/PCD 7 (Figure 5B). Prime–boost-vaccinated pigs challenged with Wa HRV had significantly higher serum IgA after challenge as compared to controls for all three P-types; these titers were comparable again to trivalent nanoparticle 2×-vaccinated pigs (Figure 5B). In the Arg HRV-challenged group, P[8]-specific serum IgA titers were significantly higher in both vaccine groups as compared to controls after two vaccinations (Figure 6B). However, after challenge, prime–boost pigs had significantly higher P[8]-specific serum IgA titers than controls, with trivalent nanoparticle 2× group having IgA titers only slightly higher than the controls at this timepoint (Figure 6B). Both vaccine groups in the Arg HRV-challenged group had significantly higher P[6] and P[4]-specific serum IgA titers on PVD 28, which lasted until PVD 35/PCD 7 (Figure 6B).

### 3.3. Prime–Boost Vaccinated Gn Pigs Challenged with Arg HRV Had Significantly Higher P[8]-Specific IgA in the Small Intestine at Euthanasia

At PVD 35/PCD 7, Arg HRV-challenged pigs who received the prime–boost regimen had significantly higher P[8]-specific IgA antibody titers in SIC as compared to other groups (Figure 7B). Wa HRV-challenged pigs had substantially higher titers of P[8]-specific IgA, but they were not statistically significant (Figure 7A). Arg HRV-challenged pigs who received the prime–boost regimen also had higher titers of P[6]-specific IgA in the SIC, but they were not statistically significant either (Figure 7B).

### 3.4. Both Vaccine Regimens Induced Strong P[8]- and P[6]-Specific Virus Neutralizing Antibody Responses in the Serum of Wa and Arg HRV-Challenged Gn Pigs

Both vaccine groups in the Wa HRV-challenged group had significantly higher P[8]-specific VN titers at PVDs 14 and 28 (Figure 8A). By PVD 35/PCD 7, there were no significant differences among the three groups, though mean P[8]-specific VN titers were visibly higher in prime–boost vaccinated pigs challenged with Wa HRV. P[6]-specific VN titers were significantly higher in Wa HRV-challenged trivalent nanoparticle 2×-vaccinated pigs after two vaccinations at PID 28, and both vaccine groups had significantly higher mean P[6]-specific VN titers at PID 35 as compared to controls (Figure 8A). There were no significant differences in P[4]-specific VN titers among Wa HRV-challenged groups at any timepoint (Figure 8A). P[8]-specific VN titers were higher in both vaccine groups challenged with Arg HRV after two vaccinations and post-challenge, but the differences were not significant (Figure 8B). P[6]-specific VN titers were significantly higher in trivalent nanoparticle 2×-vaccinated Gn pigs challenged with Arg HRV at PID 28, but by PID 35/PCD 7, there were no differences between groups (Figure 8B). Arg HRV-challenged Gn pigs vaccinated with the prime–boost regimen exhibited higher mean P[4]-specific VN titers at PID 35/PCD 7, but like the Wa HRV-challenged pigs, they were not statistically significant (Figure 8B).

### 3.5. Trivalent Nanoparticle 2×-Vaccinated, Wa HRV-Challenged Gn Pigs Had Significantly Higher Total Numbers and Frequencies of P[8]-Specific CD3+CD4+IFN-γ+ and CD3+CD8+IFN-γ+ T Cells in the Ileum Post-Challenge

Lymphocytes extracted from peripheral blood, ileum, and spleen were restimulated in vitro with P[8], P[6], or P[4] VP8* proteins and analyzed for their subsequent intracellular production of IFN-γ. Trivalent nanoparticle 2×-vaccinated pigs challenged with Wa HRV had significantly higher total numbers of both CD3+CD4+IFN-γ+ and CD3+CD8+IFN-γ+ T cells, as well as significantly higher frequencies of P[8]-specific CD3+CD4+IFN-γ+ T cells in the ileum compared to the control pigs (Figure 9A,B). These T cell responses were also increased in the spleen and blood, but not significantly so (Figure 9A,B). Total numbers and mean frequencies of both P[6]-specific CD3+CD4+IFN-γ+ and CD3+CD8+IFN-γ+ T cells were higher in all tissues of Wa HRV-challenged pigs vaccinated with the trivalent nanoparticle 2× regimen than the controls, but not to a significant degree (Figure 9A,B). Total numbers and mean frequencies of both P[4]-specific CD3+CD4+IFN-γ+ and CD3+CD8+IFN-γ+ T cells were higher in the ileum of Wa HRV-challenged pigs vaccinated with trivalent nanoparticle 2×, but, again, not significantly so (Figure 9A,B). In Arg HRV-challenged pigs, trivalent nanoparticle 2× vaccination also primed for higher total numbers and mean frequencies of P[8]-specific CD3+CD8+IFN-γ+ T cells in all tissues, and CD3+CD4+IFN-γ+ T cells in the ileum and spleen post-challenge, though these values were not statistically significant (Figure 10A,B). P[6]-specific total numbers and mean frequencies of both CD3+CD4+IFN-γ+ and CD3+CD8+IFN-γ+ T cells were elevated in all tissues of trivalent nanoparticle 2×-vaccinated pigs challenged with Arg HRV as compared to controls, except for P[6]-specific CD3+CD4+IFN-γ+ mean frequencies in the blood, which were similar to controls (Figure 10A,B). The T cell responses were unfortunately not determined for the prime–boost vaccinated pigs.

### 3.6. Both Vaccine Regimens in Both Challenge Groups Induced Significant P-Type-Specific ASC Responses in the Ileum and Spleen

P[8], P[6], and P[4]-specific IgG and IgA ASC responses in the ileum, spleen, and blood of all Gn pigs were evaluated at PID 35/PCD 7. Significantly, ASC responses in the ileum and spleen were primed by both vaccine regimens. In Wa HRV-challenged pigs vaccinated with the trivalent nanoparticle 2× regimen, significantly higher numbers of P[8]- and P[4]-specific IgG ASCs were detected in the ileum compared to the control pigs (Figure 11A). Furthermore, for pigs in this challenge group vaccinated with the prime–boost regimen, P[8]-specific IgG ASC numbers in the spleen were significantly higher than the controls (Figure 11A). In Arg HRV-challenged pigs, P[8]- and P[6]-specific IgG ASC numbers were significantly higher in the ileum of prime–boost vaccinated pigs (Figure 11B). These pigs also had significantly higher numbers of P[8]-specific IgA ASCs in the spleen (Figure 11B). Additionally, P[8]- and P[6]-specific IgG ASCs of trivalent nanoparticle 2×-vaccinated pigs in the ileum were also higher or significantly higher than the controls (Figure 11B). The ASC responses in tissues and vaccine/challenge groups that are not presented here did not differ significantly compared to the control group.

## 4. Discussion

In this study, we evaluated the immunogenicity and protective efficacy of a trivalent nanoparticle vaccine in two different dosing regimens using Gn pig models of Wa (G1P[8]) and Arg (G4P[6]) HRV infection and diarrhea. The first regimen consisted of two IM administrations, and the other an oral priming with a live oral attenuated HRV vaccine followed by one IM boosting with the trivalent nanoparticle vaccine. We found that neither regimen prevented the incidence of diarrhea upon challenge with 1 × 10^5^ FFU of the virulent Wa or Arg HRV. However, there was a significant reduction of virus shedding in both challenge groups vaccinated with the prime–boost regimen, as evidenced by the delayed onset and the significant reduced duration of virus shedding for Wa P[8]-challenged pigs and reduced duration, mean peak titers, and AUC of virus shedding for Arg P[6]-challenged pigs. Furthermore, the trivalent nanoparticle 2× regimen also significantly reduced virus shedding in Arg P[6] HRV-challenged pigs, as evidenced by the significantly reduced mean peak titers and AUC. These results agree with previous studies in mice, where mice vaccinated with S_60_ particles fused with murine rotavirus (EDIM) VP8* had significantly reduced viral antigen shedding after homotypic challenge as compared to controls [33]. Our current studies of Gn pigs and studies of mice have also similarly demonstrated the robust immunogenicity of the nanoparticle vaccine platform delivered IM (or subcutaneously) [33,34].

Despite the presence of serum IgA, IgG, and VN antibodies against both P-types of the HRV challenge strains at the time of challenge (PVD28/PCD0), trivalent nanoparticle 2×-vaccinated pigs were not protected against diarrhea. A previous study evaluating a similar parenterally administered P24-VP8* vaccine with three IM doses (200 µg/dose) in Gn pigs showed significant protection against diarrhea—shortening the duration from 6.0 to 3.3 days and mean diarrhea scores from 14.3 to 9.1—despite a very similar degree of protection against virus shedding—shortening duration of virus shedding from 5.9 to 2.5 days [23]. Based on the encouraging testing results in mice [33,34], we were hoping that two doses of S60-VP8* or one Rotarix oral priming and one IM S60-VP8* boosting were sufficient. Only one booster and the actual lower doses (due to production sterility issues) in this study may have greatly contributed to the lack of protection against diarrhea by the trivalent nanoparticle 2× regimen. Significantly higher P[8]- and P[6]-specific, but not P[4]-specific, serum VN antibody titers were induced by both vaccine regimens. It is plausible that the P[4]-VP8* component of the trivalent vaccine was less immunogenic due to antigen quality issues (i.e., lower concentration, protein degradation). Further analysis is needed to identify the cause.

In our study, Gn pigs who received an oral priming dose of Rotarix^®^ and an IM boosting dose of S_60_-VP8* had significantly reduced diarrhea scores from PCD 4–6 after Wa HRV challenge. Intestinal immune responses, namely HRV-specific IgA and IFN-γ producing CD4+ and CD8+ T cells induced by vaccination or natural infection are extremely important in the protection from clinical signs of subsequent HRV infection. Without these immune effectors at the site of viral entry and replication, cells are susceptible to HRV infection, resulting in the various cellular mechanisms that lead to HRV-associated diarrhea. The observed reduction in diarrhea scores is likely due to the priming dose of the replicating vaccine-induced intestinal IgA and effector T cells, which can prevent or reduce infection, in combination with circulating antibody responses, which are responsible for the reduction of viremia to reduce the perpetuation of HRV infection in the intestine [41].

Both challenge groups vaccinated with both regimens showed some cross-P-type induction of IgG ASCs in the spleen and ileum. Virus-specific IgG ASCs are not historically correlated with protection against HRV disease, while intestinal IgA ASCs are reported to be a correlate of protection [37]. The lack of intestinal IgA ASCs may also partially explain the lack of protection against diarrhea. HRV-specific IFN-γ+ T cell responses were induced by the trivalent nanoparticle 2× regimen in Wa HRV-challenged pigs, with P[8]-specific responses being significantly higher in the ileum and substantially higher in all other tissues, including P[6]- and P[4]-specific, than the controls. The effector T cell responses should have contributed to the reduction of virus shedding. Previously, it was thought that T cell responses, particularly in the mucosa, were difficult to induce through IM vaccination alone; however, this paradigm appears to be shifting [42] and is further evidenced by the current study. Furthermore, studies have suggested that the site of antigen entry determines T cell homing capacities [43,44]. In this case, oral immunization would likely induce T cells’ acquisition of homing receptors specific to the gut, whereas parenteral immunization would allow for homing capacities to multiple systemic sites [43]. This would explain the high numbers of HRV-specific T cells in both the ileum, blood, and spleen of the trivalent nanoparticle 2×-vaccinated pigs. We would expect similar if not greater responses to be induced in the prime–boost vaccinated pigs. Further studies are needed to demonstrate the promise of an oral prime/parenteral boost HRV vaccine regimen, as HRV-specific IFN-γ+ T cells in the gut of Gn pigs are strongly associated with protection from challenge [38]. This, along with protection offered by systemic HRV-specific IFN-γ+ T cell responses and antibody/ASC responses, is likely to be highly effective at suppressing primary viremia and intestinal replication of HRV during infection [41,45,46].

## 5. Conclusions

The S_60_-VP8* nanoparticle vaccine candidate is immunogenic and conferred partial protection against virus shedding in Gn pig models of P[6] and P[8] HRV infection both as a two-dose parenteral regimen, as well as in combination with a live oral priming dose of attenuated HRV vaccine. These results are echoed by mouse model studies that have also shown high immunogenicity for all three P-types included in the formulation [34]. However, three doses of vaccination are likely needed for the vaccine to realize its potential effectiveness against HRV diarrhea. Our findings support the continued investigation of parenteral nanoparticle-based HRV vaccines in a primary and booster vaccine regimen for the improvement of efficacy in areas of the world where HRV is still resulting in significant morbidity and mortality.

## Figures and Tables

**Figure 1 vaccines-11-00927-f001:**
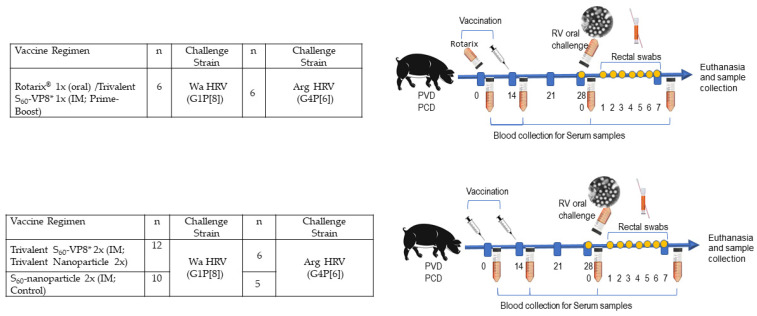
Study design to assess the immune response and protective efficacy of the vaccine regimens in Gn pigs. Pigs were vaccinated with two doses of vaccines on PVD 0 and 14. Blood/serum samples were collected on PVD 0, 14, 28, and PCD 7. The immunized pigs were challenged with virulent Wa HRV or Arg HRV on PCD 0 and rectal swab samples were collected daily post-challenge for seven days for the assessment of diarrhea and virus shedding.

**Figure 2 vaccines-11-00927-f002:**
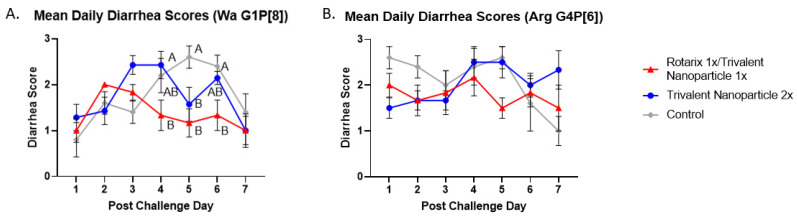
**Daily mean diarrhea scores from PCD 0–7 in Wa (A) and Arg HRV-challenged (B) Gn pigs.** At PVD 28, Gn pigs were orally challenged with 1 × 10^5^ FFU of either Wa or Arg HRV and monitored for diarrhea severity and duration via daily rectal swabs for 7 days. Differing letters indicate significant difference (two-way ANOVA of repeated measures through time, followed by Tukey’s multiple comparisons test; *p* ≤ 0.05), and error bars indicate SEM.

**Figure 3 vaccines-11-00927-f003:**
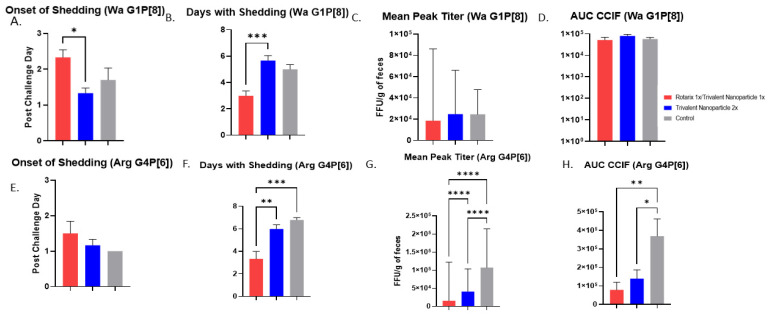
**Protection from virus shedding as measured by CCIF in Wa- and Arg HRV-challenged Gn pigs.** At PVD 28, pigs were orally challenged with 1 × 10^5^ FFU of either Wa or Arg HRV and monitored for virus shedding via daily rectal swabs for 7 days. (**A**,**E**) Mean onset day of virus shedding in Wa or Arg HRV-challenged pigs; (**B**,**F**) Mean days with virus shedding in Wa or Arg HRV-challenged pigs; (**C**,**G**) Mean peak virus shedding titers in Wa or Arg HRV-challenged pigs; (**D**,**H**) Mean area under the curve of virus shedding in Wa or Arg HRV-challenged pigs. Asterisks indicate degree of statistically significant difference (ordinary one-way ANOVA; * *p* ≤ 0.05, ** *p* ≤ 0.01, *** *p* ≤ 0.001, **** *p* ≤ 0.0001). Error bars indicate SEM or (geometric SD for peak titers).

**Figure 4 vaccines-11-00927-f004:**
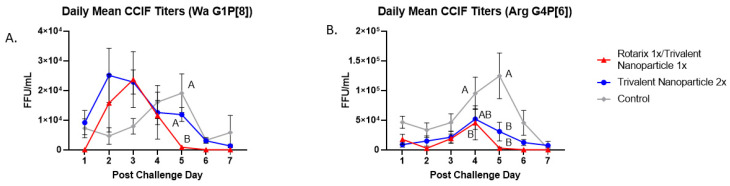
**Daily fecal virus shedding as measured by CCIF in prime-boost, trivalent 2x and control-vaccinated pigs from PCDs 0-7.** Daily rectal swabs were taken for evaluating fecal virus shedding after challenge with Wa HRV. Infectious virus particles were measured by CCIF and results are expressed as FFU/mL. Fecal samples from mock-infected pigs were used as negative controls. Different capital letters above data points indicate significant differences between groups at the same time point, while shared letters or no letters indicate no significant difference, according to two-way ANOVA of repeated measures through time, followed by Tukey’s multiple comparisons test (*p* ≤ 0.05). Bars indicate SEM. CCIF, cell culture immunofluorescence; FFU, focus forming units. (**A**) Daily Mean CCIF Titers (Wa G1P[8]), (**B**) Daily Mean CCIF Titers (Arg G4P[6]).

**Figure 5 vaccines-11-00927-f005:**
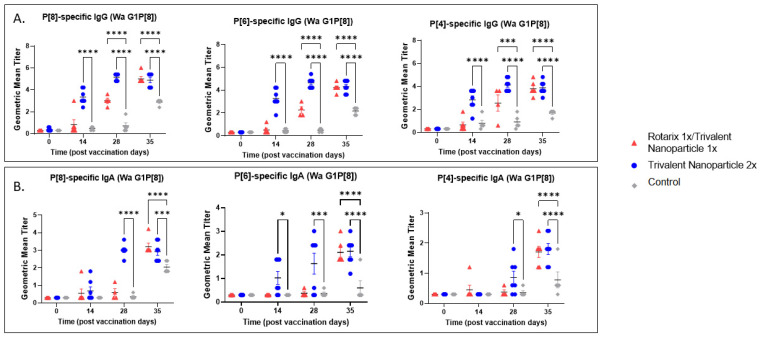
**P[8], P[6], and P[4]-specific serum antibody responses in Wa HRV-challenged Gn pigs.** Serum was collected at PVD 0, 14, 28 (PCD 0), and 35 (PCD 7) to evaluate systemic P-type-specific IgG (**A**) and IgA (**B**) responses using direct ELISA. Asterisks indicate degree of statistically significant difference (mixed-effects analysis with Dunnett’s multiple comparisons test; * *p* ≤ 0.05, *** *p* ≤ 0.001, **** *p* ≤ 0.0001). Lines and error bars indicate mean and SEM, respectively.

**Figure 6 vaccines-11-00927-f006:**
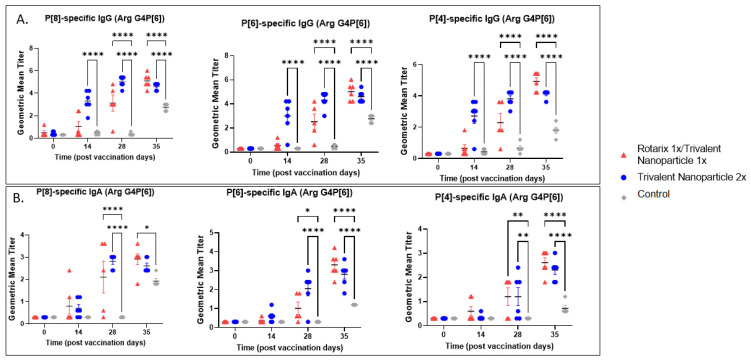
**P[8], P[6], and P[4]-specific serum antibody responses in Arg HRV-challenged Gn pigs.** Serum was collected at PVD 0, 14, 28 (PCD 0), and 35 (PCD 7) to evaluate systemic P-type-specific IgG (**A**) and IgA (**B**) responses using direct ELISA. Asterisks indicate degree of statistically significant difference (mixed-effects analysis with Dunnett’s multiple comparisons test; * *p* ≤ 0.05, ** *p* ≤ 0.01, **** *p* ≤ 0.0001). Lines and error bars indicate mean and SEM, respectively.

**Figure 7 vaccines-11-00927-f007:**
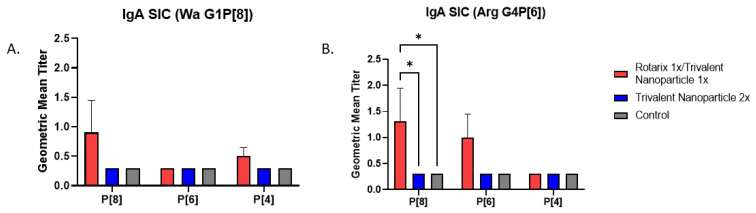
Intestinal IgA antibody titers in vaccinated and control Gn pigs challenged with either Wa (**A**) or Arg HRV (**B**). At PVD 35/PCD 7, Gn pigs were euthanized and small intestinal contents were collected and evaluated using direct ELISA to detect P-type specific IgA antibody responses. Asterisks indicate degree of statistically significant difference (mixed-effects analysis with Tukey’s multiple comparisons test; * *p* ≤ 0.05). Error bars indicate SEM.

**Figure 8 vaccines-11-00927-f008:**
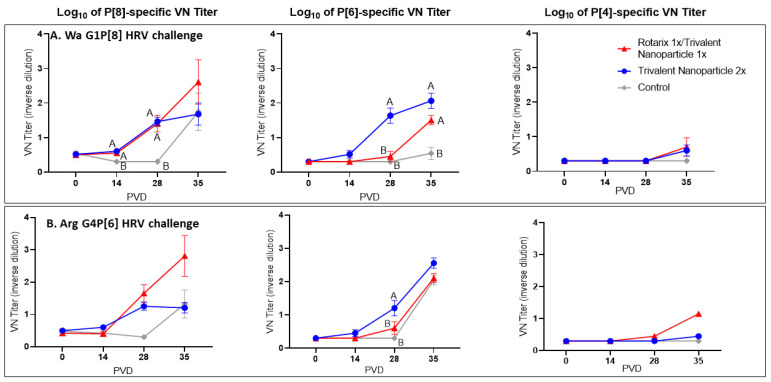
**Serum virus neutralization of P[8], P[6], and P[4] HRV in Wa HRV (A) or Arg HRV-challenged Gn pigs (B).** Serum was collected at PVD 0, 14, 28 (PCD 0), and 35 (PCD 7) and evaluated for neutralizing antibody activity. Differing letters indicate significant difference (mixed-effects analysis using two-stage linear step-up procedure, *p* ≤ 0.05), and error bars indicate SEM.

**Figure 9 vaccines-11-00927-f009:**
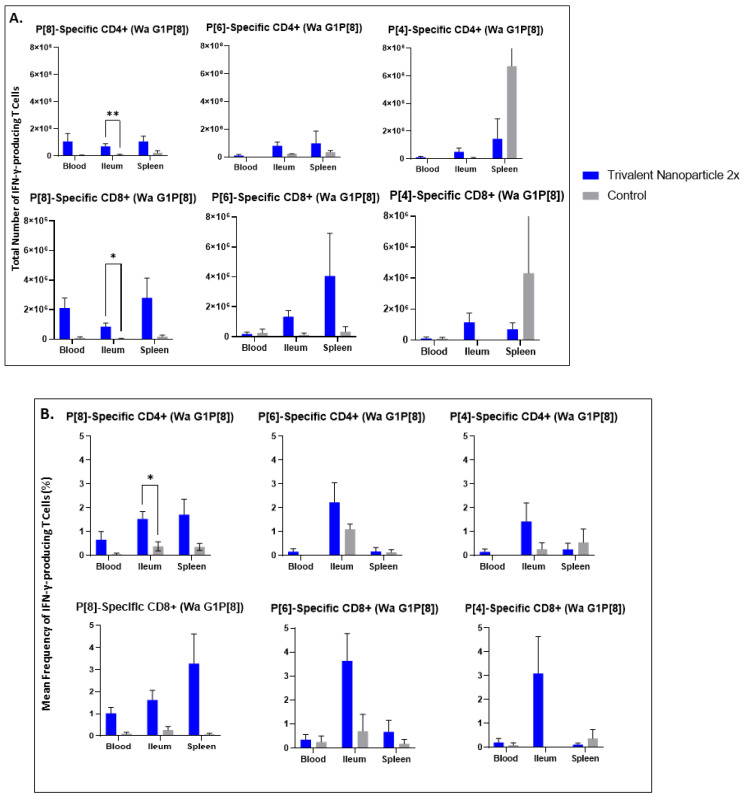
Total numbers (**A**) and frequencies (**B**) of P[8], P[6] and P[4]-specific IFN-γ-producing CD3+CD4+ and CD3+CD8+ T cells in blood, ileum and spleen of Wa HRV-challenged Gn pigs at PCD 7 detected by flow cytometry. MNCs were stimulated in vitro for 17h with P[8], P[6] and P[4] S60-VP8* antigen, or mock stimulated. Asterisks indicate degree of statistically significant difference (multiple Mann-Whitney tests; * *p* ≤ 0.05, ** *p* ≤ 0.01). Error bars indicate SEM.

**Figure 10 vaccines-11-00927-f010:**
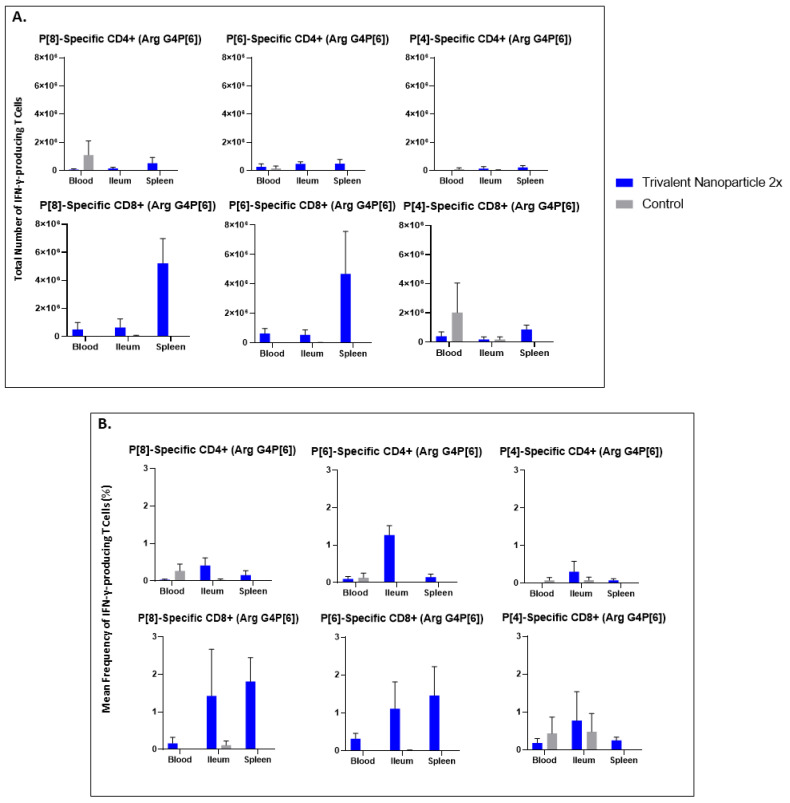
**Total numbers (A) and frequencies (B) of P[8], P[6] and P[4] type-specific IFN-γ-producing CD3+CD4+ and CD3+CD8+ T cells in blood, ileum and spleen of Arg HRV challenged Gn pigs at PCD 7 detected by flow cytometry.** MNCs were stimulated in vitro for 17 h with P[8], P[6] and P[4] S60-VP8* antigen, or mock stimulated. Asterisks indicate degree of statistically significant difference (multiple Mann-Whitney tests). Error bars indicate SEM.

**Figure 11 vaccines-11-00927-f011:**
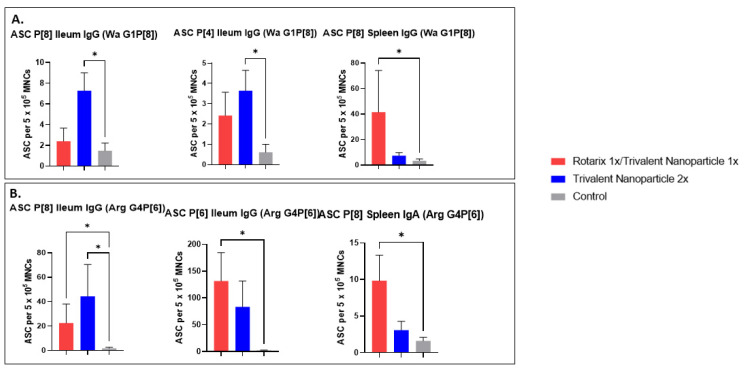
P-type specific antibody-secreting cell (ASC) responses in the ileum and spleen of Wa (**A**) or Arg (**B**) HRV-challenged Gn pigs at PCD 7. Asterisks indicate degree of statistically significant difference (Kruskal-Wallis test with Dunn’s multiple comparisons test; * *p* ≤ 0.05). Error bars indicate SEM. Only ASC responses with significant differences among the groups are presented.

**Table 1 vaccines-11-00927-t001:** Diarrhea and virus shedding in vaccinated and control pigs after challenge with virulent Wa HRV (G1P[8]).

Clinical Signs of Diarrhea ^b^							Virus Shedding (CCIF) ^b^				
Treatments (Vaccine Dose µg) ^a^	n	Percentage with Diarrhea	Mean Days to Onset	Mean Duration Days	Mean Cumulative Fecal Score	AUC of Diarrhea	Percentage of Shedding Virus	Mean Days to Onset	Mean Duration Days	Mean Peak Titer (FFU/g of Feces) ^e^	AUC of Virus Shedding ^e^
Rotarix 1x/Trivalent Nanoparticle 1x	6	100%	2	3.70	10.3	8.70	100%	2.33 (0.21) ^A,c,d^	3.00 (0.4) ^B^	18,594	51,800
Trivalent Nanoparticle 2x	12	100%	2	4.58	12.4	11.1	100%	1.33 (0.14) ^B^	5.67 (0.38) ^A^	24,771	80,583
Control	10	100%	2	4.86	13.0	11.4	100%	1.70 (0.33) ^AB^	5.00 (0.37) ^A^	24,552	57,440

Note: a. Pigs were immunized two times with S_60_ nanoparticle control, trivalent nanoparticle vaccine, or once with Rotarix followed by trivalent nanoparticle or control, at 5 (post-vaccination day [PVD] 0), and 19 days (PVD 14) of age. On PVD 28, all pigs were orally challenged with 1 × 10^5^ FFU of virulent Wa HRV and monitored for diarrhea and virus shedding for 7 days post-challenge. b. Fecal consistency scores were used to assess diarrhea; scores are defined as 0: solid, 1: pasty, 2: semi-liquid, and 3: liquid. Scores of 2 or higher are considered diarrheic. Rotavirus shedding titers were determined by rotavirus antigen ELISA (detect viral antigen) and CCIF (determine the number of infectious viral particles). If there is no diarrhea or virus shedding, the mean days to onset were assigned as one day after the pigs were euthanized (8) for statistical analysis. c. Different letters indicate significant differences between groups (n = 6–10; *p* ≤ 0.05), while shared letters or no letters indicate no significant difference. d. Numbers in parentheses represent the standard error of the mean (SEM). e. Mean peak titer and AUC of virus shedding titers were lgo10 transformed for statistical analysis.

**Table 2 vaccines-11-00927-t002:** Diarrhea and virus shedding in vaccinated and control pigs after challenge with virulent Arg HRV (G4P[6]).

Clinical Signs of Diarrhea ^b^							Virus Shedding (CCIF) ^b^				
Treatments (Vaccine Dose µg) ^a^	n	Percentage with Diarrhea	Mean Days to Onset	Mean Duration Days	Mean Cumulative Fecal Score	AUC of Diarrhea	Percentage of Shedding Virus	Mean Days to Onset	Mean Duration Days	Mean Peak Titer (FFU/g of Feces) ^e^	AUC of Virus Shedding ^e^
Rotarix 1x/Trivalent Nanoparticle 1x	6	100%	2	4.70	13.0	10.8	100%	1.50	3.33 (0.67) ^B,c,d^	15,554 (26,842) ^C^	78,200 (41,088) ^B^
Trivalent Nanoparticle 2x	6	100%	2	5.00	14.2	12.3	100%	1.17	6.00 (0.37) ^A^	40,778 (15,756) ^B^	138,767 (47,474) ^B^
Control	5	100%	1	5.20	14.6	12.8	100%	1.00	6.80 (0.20) ^A^	107,187 (36,620) ^A^	368,600 (93,322) ^A^

Note: a. Pigs were immunized two times with S_60_ nanoparticle control, trivalent nanoparticle vaccine, or once with Rotarix followed by trivalent nanoparticle or control, at 5 (post-vaccination day [PVD] 0), and 19 days (PVD 14) of age. On PVD 28, all pigs were orally challenged with 1 × 10^5^ FFU of virulent Arg HRV and monitored for diarrhea and virus shedding for 7 days post-challenge. b. Fecal consistency scores were used to assess diarrhea; scores are defined as 0: solid, 1: pasty, 2: semi-liquid, and 3: liquid. Scores of 2 or higher are considered diarrheic. Rotavirus shedding titers were determined by rotavirus antigen ELISA (detect viral antigen) and CCIF (determine the number of infectious viral particles). If there is no diarrhea or virus shedding, the mean days to onset were assigned as one day after the pigs were euthanized (8) for statistical analysis. c. Different letters indicate significant differences between groups (n = 6–10; *p* ≤ 0.05), while shared letters or no letters indicate no significant difference. d. Numbers in parentheses represent the standard error of the mean (SEM). e. Mean peak titer and AUC of virus shedding titers were lgo10 transformed for statistical analysis.

## Data Availability

All relevant data are included within the manuscript.

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
