# Peer review of "Combined Live Oral Priming and Intramuscular Boosting Regimen with Rotarix® and a Nanoparticle-Based Trivalent Rotavirus Vaccine Evaluated in Gnotobiotic Pig Models of G4P[6] and G1P[8] Human Rotavirus Infection"

_vaccines, 2023, doi:10.3390/vaccines11050927_

Round 1

Reviewer 1 Report

The manuscript by Hensley et al. describes the results of testing two strategies for anti-rotaviral vaccination in gnotobiotic pig model. It is a limited, but helpful study providing evidence that even a reasonably efficient immune response to rotaviral vaccination may not provide a tangible clinical benefit. There are several improvements that authors need to make prior to their manuscript being accepted.

1) Lack of vaccine efficacy    in medium-to-low-income countries claimed by the authors needs to be better explained, especially in the Abstract, but also in Introduction (lines 43-46). Note also a typo on line 45 (an extra 's' in LMICSs).

2) It doesn't look that ' 45 pigs were assigned randomly to three groups per challenge strain' (line 139) since there were only two challenge strains; the way data is described there were three test groups (two vaccinated and one control).

3) There is absolutely no need to show error bars as SEM, this is simply incorrect, even if it looks better than SD. Please show all the error bars as SD.

4) The legend to Gig. 2 says that "different letters indicate significant difference," but it is impossible to understand which group is different from each one (and the graph itself shows a very little difference, if any, to be perfectly honest).

5) Why there are no error bars in Fig. 3G?

6) Differences in immune responses in immunized groups vs. non-immunized prior to the challenge (up to d28) don't tell us much, they are supposed to be different. Moreover, this data calls in to question a very early time-point for viral challenge selected by authors. It is not unlikely that this has contributed to overall protection failure that they have observed.

7) Furthermore, this failure needs to be spelled out in the Abstract (it is not) and discussed in some detail throughout the Discussion since this is the main conclusion that can be drawn from this study. Which immune correlates authors think to be the most helpful or telling? Which regimens or vaccine formulations they suggest to use in the future? Why did they decide to give less doses (minus one booster) and a smaller dose on top of it? What experimental path they suggest to follow?

Author Response

Responses to reviewer 1 comments.

1) Lack of vaccine efficacy in medium-to-low-income countries claimed by the authors needs to be better explained, especially in the Abstract, but also in Introduction (lines 43-46). Note also a typo on line 45 (an extra 's' in LMICSs).

--Lack of efficacy in LMICs has been noted in the abstract (within the word limit allowance), and expanded upon in the introduction (lines 43-51) to read as “These vaccines, though highly effective in developed countries, do not offer the same protection for children in LMICs for various reasons, mainly associated with the gastrointestinal (GI) tract [1, 2]. Some of these issues include malnutrition, circulating maternal antibodies, concurrent use of other oral vaccines, and gut dysbiosis [3]. Malnutrition and gut dysbiosis have been associated with reduced immune response to live oral attenuated rotavirus vaccines previously [3]. Furthermore, there is evidence that the use of live oral poliovirus vaccines, which are often given in conjunction with HRV vaccines, can inhibit HRV vaccine replication, further reducing vaccine take [3].”

The typo from line 45 (LMICSs) has been corrected.

2) It doesn't look that ' 45 pigs were assigned randomly to three groups per challenge strain' (line 139) since there were only two challenge strains; the way data is described there were three test groups (two vaccinated and one control).

--We reworded line 139 to reflect there were 45 pigs total used for the entire study (across 3 groups per challenge strain) to read as “Pigs were inoculated with either the priming dose of trivalent S60-VP8* IM (group 1) or Rotarix® orally on post-partum day (PPD) 5 (also referred to as post-vaccination day [PVD] 0) and subsequently received the same dose IM boosting of trivalent S60-VP8* at PVD 14 (group 2) (Figure 1). Control pigs (group 3) were immunized on the same schedule with two IM administrations of the S60 nanoparticle without VP8* antigen.”

3) There is absolutely no need to show error bars as SEM, this is simply incorrect, even if it looks better than SD. Please show all the error bars as SD.

--We tried to show SD on figures, but they make the figures too crowed to read. We have the options of not showing error bars or show SEM as we did. Since the proper statistical analyses have been performed and when significances are found, we indicated them in the figures. So, we would like not to make changes to the figures.

4) The legend to Gig. 2 says that "different letters indicate significant difference," but it is impossible to understand which group is different from each one (and the graph itself shows a very little difference, if any, to be perfectly honest).

--We revised figure legends to clarify the use of differing letter for statistical significance to read as “Different letters (A, B) indicate significant differences between groups (n = 6-12; p ≤  0.05), while shared letters (A, AB) or no letters indicate no significant difference.”

5) Why there are no error bars in Fig. 3G?

--We added SD bars to Figure 5G.

6) Differences in immune responses in immunized groups vs. non-immunized prior to the challenge (up to d28) don't tell us much, they are supposed to be different. Moreover, this data calls in to question a very early time-point for viral challenge selected by authors. It is not unlikely that this has contributed to overall protection failure that they have observed.

-- The different regimens’ pre-challenge immune responses data was expected to potentially differ significantly and they give information about the priming ability of the different vaccine regimens. The timepoint of virus challenge was used in accordance with previously published studies that use the well-established neonatal Gn pig model of HRV infection and diarrhea for preclinical vaccine evaluation.  The Gn pig model with the same time-point for viral challenge has been used to demonstrate strong vaccine protection in several previous studies, so the time-point is not considered a contributing factor for the overall protection failure we observed in this study.

7) Furthermore, this failure needs to be spelled out in the Abstract (it is not) and discussed in some detail throughout the Discussion since this is the main conclusion that can be drawn from this study.

--Failure to protect from disease has been added in the Abstract to read as “The two vaccine regimens failed to confer significant protection against diarrhea” and discussed in some detail in the Discussion.

Which immune correlates authors think to be the most helpful or telling?

-- Based on our previous studies, HRV-specific IgA ASC and IFN-g producing CD4+ and CD8+ T cells induced by vaccination in the small intestine present at the time of virus challenge (PCD 0) are the most important immune correlates of protection.  In this current study, we could not collect intestinal tissues at PCD 0 to measure these immune responses. As we observed in this study and also previous studies, the immune responses measured at PCD 7 are mixtures of responses to vaccination and virus infection, they do not correlate well with protection. Therefore, no clear conclusion can be made based the data from this study.

Which regimens or vaccine formulations they suggest to use in the future?

--We noted in the Conclusions that “Our findings support the continued investigation of parenteral nanoparticle-based HRV vaccines in a primary and booster vaccine regimen for the improvement of efficacy”.

Why did they decide to give less doses (minus one booster) and a smaller dose on top of it?

-- We revised the sentences in the Discussion to explain “Based on the encouraging testing results in mice [23, 24], we were hoping that two doses of S60-VP8* or one Rotarix oral priming and one IM S60-VP8* boosting were sufficient. Only one booster and the actual lower doses (due to production sterility issue) in this study may have greatly contributed to the lack of protection against diarrhea by the trivalent nanoparticle 2x regimen.”

What experimental path they suggest to follow?

--We suggested future development of the prime/boost regimen in the Conclusion (both as a result of the data from the current study and historical data).

Reviewer 2 Report

Combined live oral priming and intramuscular boosting regimen with Rotarix® and a nanoparticle-based trivalent rotavirus vaccine evaluated in gnotobiotic pig models of G4P[6] and G1P[8] human rotavirus infection.

Manuscript ID: vaccines-2346986.

In this study, the authors developed a two-dose intramuscular (IM) regimen of the trivalent, nanoparticle-based, nonreplicating, HRV vaccine (trivalent S60-VP8*), utilizing the shell (S) domain of the capsid of norovirus (NoV) as an HRV VP8* antigen display platform, was evaluated for safety, immunogenicity and protective efficacy against P[6] and P[8] HRV using Gn pig models. A prime-boost strategy using one dose of the commercial oral Rotarix® vaccine, fol-lowed by one dose of the IM trivalent nanoparticle vaccine was also evaluated. Both regimens were highly immunogenic in terms of inducing serum virus neutralizing (VN), IgG, and IgA an-tibodies. The prime-boost regimen significantly shortened the duration of virus shedding in pigs challenged orally with the virulent Wa (G1P[8]) HRV and significantly shortened the mean duration of virus shedding, mean peak titer, and area under the curve (AUC) of virus shedding after challenge with Arg (G4P[6]) HRV. Prime-boost-vaccinated pigs challenged with P[8] HRV had significantly higher P[8]-specific IgG antibody-secreting cells (ASCs) in the spleen post-challenge. 

The study is well-designed and perfectly executed. But the figures are missing. Can you add the figures for the result section and some schematics as well.

No comments

Author Response

--We thank the reviewer for the compliments.

--The tables and figures are all in the attached file.

Round 2

Reviewer 2 Report

The manuscript looks perfect in its present form and it can be accepted in this journal. The dtudy is very interesting and well described. The authors have improved the manuscript and incorporated figures and tables. 

None